# Effectiveness of Manual Trigger Point Therapy in Patients with Myofascial Trigger Points in the Orofacial Region—A Systematic Review

**DOI:** 10.3390/life13020336

**Published:** 2023-01-27

**Authors:** Frauke Müggenborg, Ester Moreira de Castro Carletti, Liz Dennett, Ana Izabela Sobral de Oliveira-Souza, Norazlin Mohamad, Gunnar Licht, Harry von Piekartz, Susan Armijo-Olivo

**Affiliations:** 1Department of Physiotherapy, University of Applied Sciences Osnabrück, Faculty of Economics and Social Sciences Caprivistr. 30A, 49076 Osnabrück, Germany; 2Post Graduate Program in Human Movement Sciences, Methodist University of Piracicaba-UNIMEP, Piracicaba 13400-911, Brazil; 3Scott Health Sciences Library, University of Alberta, Edmonton, AB T6G 1C9, Canada; 4Graduate Program in Neuropsychiatry and Behavioral Sciences, Federal University of Pernambuco (UFPE), Av. Prof. Moraes Rego, 1235, Recife 50670-901, Brazil; 5Department of Physical Therapy, Faculty of Rehabilitation Medicine, Rehabilitation Research Center, University of Alberta, Edmonton, AB T6G 1C9, Canada; 6Centre of Physiotherapy, Faculty of Health Sciences, Universiti Teknologi MARA, Puncak Alam Campus, Shah Alam 42300, Malaysia; 7FOURBs-Specialist Medical Center for Orthopedics and Rehabilitation of the Locomotor System–Johannisstr. 19, 49074 Osnabrück, Germany; 8Faculty of Medicine and Dentistry, Department of Dentistry, University of Alberta, Edmonton, AB T6G 1C9, Canada

**Keywords:** trigger points, myofascial pain, manual therapy, temporomandibular disorders, systematic review

## Abstract

The objective was to compile, synthetize, and evaluate the quality of the evidence from randomized controlled trials (RCTs) regarding the effectiveness of manual trigger point therapy in the orofacial area in patients with or without orofacial pain. This project was registered in PROSPERO and follows the PRISMA guidelines. Searches (20 April 2021) were conducted in six databases for RCTs involving adults with active or latent myofascial trigger points (mTrPs) in the orofacial area. The data were extracted by two independent assessors. Four studies were included. According to the GRADE approach, the overall quality/certainty of the evidence was very low due to the high risk of bias of the studies included. Manual trigger point therapy showed no clear advantage over other conservative treatments. However, it was found to be an equally effective and safe therapy for individuals with myofascial trigger points in the orofacial region and better than control groups. This systematic review revealed a limited number of RCTs conducted with patients with mTrPs in the orofacial area and the methodological limitations of those RCTs. Rigorous, well-designed RCTs are still needed in this field.

## 1. Introduction

Myofascial trigger points (mTrPs) can be defined as hypersensitive or tender spots located within stretched muscle fibers (taut bands) of the skeletal muscles, which when compressed or stretched can cause referred or local pain [1]. Myofascial trigger points are associated with myofascial pain syndrome [1]. Myofascial pain can be defined as regional muscle pain that has increased pain sensitivity when palpated [2]. Myofascial pain and mTrP are commonly associated with orofacial pain and specifically associated with temporomandibular disorders (TMD), which are characterized by pain, reduced mouth opening, muscle or joint tenderness on palpation, limitation of mandibular movements, joint sounds, and otologic complaints such as tinnitus, vertigo or ear fullness among others. Orofacial pain (OFP) can be defined as pain originating below the orbitomeatal line, above the neck, and in front of the ears, including pain occurring in the mouth [3].

Pain disorders of the orofacial area are prevalent and can cause a significant personal and societal burden [4,5] The results of international epidemiological studies have shown that orofacial pain occurs in approximately 5–12% of the adult population, and young women are more affected by orofacial pain than men in a ratio of about 2:1 [5,6,7]. In fact, myofascial pain has been reported to play a major role in 45.3% of TMD diagnoses [8,9]. Myofascial pain is the second most common pain type of orofacial symptoms, and around 33% of the affected individuals have facial and masticatory musculoskeletal symptoms [9,10].

Due to the complexity of OFP and specifically TMD, their management has been interdisciplinary. Evidence-based treatments for these conditions, as stated by the guidelines of the American Academy of Orofacial Pain [10] include physical therapy (PT), patient education and self-management, behavioral therapy, pharmacologic management, orthopedic appliance therapy, dental and occlusal therapy, and surgery among others. Several systematic reviews [11,12,13,14,15,16] have looked at the effectiveness of these conservative therapies and have found them to be potentially effective at managing these disorders. However, the evidence is poor due to the high risk of bias and methodological issues in the primary studies.

Both non-invasive and invasive (such as manual trigger point therapy and dry needling) treatments exist for mTrPs. In the past few years, several studies have investigated the use of non-invasive therapies such as manual trigger point therapy [17], acupuncture [18], manual therapy [19], and laser [20], among others, to manage mTrPs. Manual trigger point therapy is a treatment method, which uses the hands of the therapist/doctor in a structured way to inactivate the mTrPs and to treat accompanying connective tissue changes, and movement restrictions [21]. The authors are not aware of any previous systematic review of the effectiveness of manual trigger point treatment, specifically in patients with myofascial trigger points, in the orofacial area. Based on our preliminary searches, previous reviews have included manual therapy in general, but they have not focused on manual trigger point therapy in particular and thus they have not exhaustively analyzed this specific literature and its effectiveness [13,14,22]. In addition, these reviews are already outdated since they were published almost seven years ago. The research in this area in the last couple of years has emphasized that there is a scarcity of available evidence in this field and therefore, there is an urgent need to fill this gap in the literature and provide focused and updated information regarding the effectiveness of manual trigger point therapy for the orofacial region [23,24,25].

Thus, the following objectives were set for this review: (1) To compile, synthesize, and evaluate the quality of the evidence from RCTs or clinical trials regarding the effectiveness of manual trigger point therapy compared with other treatment strategies, for managing mTrPs in the orofacial area in individuals with or without orofacial pain, and (2) To inform future practice and provide recommendations regarding manual trigger point therapy for people with mTrPs in the orofacial area.

## 2. Materials and Methods

This project was registered in PROSPERO (CRD42020169216) and reported based on the PRISMA guidelines [26].

**Data Searches:** This review was part of a large project looking at several interventions to manage orofacial pain; and manual trigger point therapy was one of them. The search (Appendix A) for all interventions was conducted at the same time and relevant keywords for manual trigger point therapy were included. A health sciences librarian (LD) conducted the searches in Medline (Ovid MEDLINE(R) ALL), Embase (Ovid interface), CINAHL PLUS with full text (EBSSCOhost interface), Cochrane Library Trials (Wiley Interface), Web of Science (Indexes=SCI-EXPANDED, SSCI, A&HCI, ESCI) and Scopus. The last search was conducted on 20 April 2021. The search included all relevant search terms from an earlier review [13] as well as new terms suggested by the research team. The search was limited to RCTs using a slightly modified version of Glanville et al.’s filter [27]. The date was limited to studies published after 2004 because of the earlier review [28]. No language limits were applied.

### 2.1. Eligibility Criteria

The eligibility criteria of this review used the PICOS format (population, intervention, comparison, outcome, and study design)

**Population:** This review considered studies that include adults (18+ years of age) diagnosed with active or latent TrPs in the orofacial region with or without orofacial pain. No limits were applied in terms of sex, ethnicity, or country of residence, but animal studies were excluded. Several diagnoses can be included in the umbrella term of orofacial pain (OFP), depending on the classification used. We included types of OFP diagnoses based on the International Classification of Orofacial Pain (ICOP) published in 2020 [3]: myofascial OFP (primary and secondary myofascial OFP), temporomandibular joint pain, OFP resembling presentations of primary headaches, and idiopathic OFP. A detailed description of these diagnoses can be found in Appendix B. The rest of the classifications stated in the ICOP were excluded due to the fact that they are most likely not associated with myofascial TrPs (e.g., dental pain (i.e., pulpal pain, periodontal pain, and gingival pain)). 

**Intervention(s)/Exposure(s):** The intervention of interest in this review was manual trigger point therapy which included the following: ischemic compression [29], trigger point pressure release [30], myofascial release [31], manual pressure on taut bands [32], passive stretching [33,34], manual fascial techniques [35], manual intraoral or extraoral release [32], and strain–counterstrain technique [36,37]. The description of these techniques can be found in Appendix C.

**Comparator(s)/Control:** Manual trigger point therapy was compared with any medical or physiotherapeutic technique that included, but was not limited to, dry needling, electrotherapy, laser therapy, exercises, acupuncture, ultrasound, splint management, medication, placebo, no treatment (control) or sham-therapy.

**Outcome:** The primary outcome of this review was pain intensity, which is frequently measured with the following tools: visual analogue scale (VAS), numeric rating scale (NRS), verbal rating scales (VRS), or graphical scales.

The secondary outcomes for this review were: pain pressure threshold (PPT) [38,39], maximal mouth opening (MMO), and mandibular range of motion (ROM) (right and left lateral excursion, protrusion) [40] among others. This review collected all outcomes reported by the included studies.

All time-points reported from the primary and secondary outcomes were analyzed (i.e., immediate post-treatment, short-term, intermediate-term, and long-term follow-up). 

**Studies:** This review included RCTs and controlled clinical trials (CCTs). All other types of studies were excluded. All therapeutic settings were included in this review.

### 2.2. Data Screening

Search results were compiled into an EndNote database and then imported into Covidence (www.covidence.org), which was used for the screening process. Two independent reviewers screened the titles and abstracts and full text. If disagreements occurred between the reviewers in the inclusion of an article, the reasons for the disagreement were discussed, and a consensus was reached.

### 2.3. Data Extraction

Data extraction was first performed independently by one reviewer using an electronic pilot-tested form created in Excel. A second reviewer checked the extracted information of each study. The data extraction contained qualitative and quantitative elements. The following qualitative elements were extracted: article information; main objective of the study, study design, type of interventions, study setting, population, diagnosis tools, data collection methods, RCT type, number of randomized groups; outcomes; data analysis, results, conclusions, limitations, among others. The quantitative elements for treatment effect estimates were extracted for outcomes at baseline and at different time points, including mean, standard deviation (SD), sample size, standard error (SE), and confidence intervals. Any disagreements on data extraction were resolved by consensus.

### 2.4. Risk of Bias (Quality Assessment)

Quality assessment (risk of bias—RoB) was conducted by two independent reviewers on all included studies using the new risk of bias tool (RoB2) recommended by the Cochrane Collaboration [41,42]. Two reviewers independently assessed the RoB in the primary studies [43]. For the overall assessment of the RoB for each study, studies were rated as follows: “high risk of bias” (if the study was rated high in at least one domain), “some concerns” (if the study was rated as “some concerns” in at least one domain and the other domains were low), or “low risk of bias” (if the study was rated as low risk in all individual domains). Similar decision rules have been used by previous studies when rating the overall risk of bias assessment of RCTs [44]. Disagreements in risk assessment ratings were resolved by consensus. In addition, we used a compiled set of items from seven scales used to evaluate the RoB in the physical therapy field [45]. This compiled set of items has been described previously and has been used in several systematic reviews of our team [12,13,46].

### 2.5. Data Synthesis

We summarized our findings using a narrative synthesis based on the type of intervention (e.g., ischemic compression, myofascial release, trigger point pressure release), type of diagnosis (e.g., latent or not latent masticatory mTrPs), and based on the type of outcome (e.g., pain intensity, maximum mouth opening, and pain pressure threshold). We presented the study results using evidence tables and forest plots when feasible. We used forest plots to visually show study results and direction of the treatment effects. Narrative and qualitative summaries were provided when possible. Revman 5.4 software was used to construct forest plots for all comparisons. Mean differences (MD) were used to analyze continuous outcomes and ordinal data were analyzed as continuous data. To interpret MD, the minimal important difference was used for each of the outcomes. To interpret the pain intensity, mouth opening, and tenderness, a mean difference of 1.9 cm [47], 5 mm [48], and 1.12 Kg/cm^2^ [49], respectively, were considered a clinically significant finding for these outcomes.

**Overall Quality of the Evidence:** The evidence was classified as high, moderate, low, and very low based on the GRADE approach based on the outcomes of interest [50]. The evidence was downgraded by one or two points when serious or very serious limitations, respectively, were found in the following domains: risk of bias, consistency of results, indirectness (reproducible, targeted to the population of interest), imprecision (insufficient data), or publication bias [50]. The evidence was upgraded based on three factors when applicable: large effect (up to 2 points increase), dose–response gradient (1-point increase), and plausible confounding that would change the effect.

## 3. Results

### 3.1. Study Selection

A total number of 8483 studies were found in the databases. Twenty-four studies were selected for full-text screening; however, among those, 20 studies were excluded based on the reasons described in the PRISMA flowchart (Figure 1). A detailed list of excluded studies and reasons for exclusion can be obtained from the authors upon request. Four studies were selected for data extraction and risk of bias assessment and were included in this systematic review [17,51,52,53].

### 3.2. Study Characteristics—Synthesis of Results

Table 1 summarizes the type of treatment, outcomes, results, and conclusions of each study. Regarding the diagnosis, two studies [17,53] included subjects with the diagnosis of trigger points and myogenic TMD using the RDC/TMD criteria [54] and two [51,52] included subjects with the diagnosis of latent mTrPs in the masseter muscles as stated by Simons et al. [2] but no OFP was identified by the authors. 

In terms of manual trigger points techniques, two studies [51,52] used the strain/counterstrain technique to treat latent mTrPs, and two studies [37,55] used ischemic compression to treat patients with myogenic TMD. 

The duration of the sessions and exposure times, as well as the duration of the whole treatment (in weeks) were poorly described in all studies. No adverse effects were described in any of the included studies. No other outcome was used besides the ones presented in Table 1 (i.e., pain intensity, pressure pain threshold, maximum/active mouth opening, and range of motion for active knee extension). The timing of outcome measurements was carried out immediately after each intervention in all four studies. No follow-up analysis was performed. 

The results and characteristics of the studies are shown in detail in Appendix D. Meta-analysis was not possible due to the low number of studies and heterogeneity among the protocols.

### 3.3. Pain Intensity 

**Comparison of manual trigger point therapy vs. control group**: Only one study [52] evaluated the effect of a manual trigger point technique (strain/counterstrain technique) compared to a control group. The qualitative comparison presented in the forest plot (Figure 2) and Table 2 (effect sizes), showed that the results favored the manual trigger point therapy (strain/counterstrain technique) when compared to the control group (MD [95% CI] 1.60 cm [0.77, 2.43]). The MD between groups was potentially clinically significant based on the minimally important difference of pain intensity reported (Figure 2) [47].

**Comparison of manual trigger point therapy vs. other therapies:** Three of the studies [17,52,53] evaluated the effect of manual trigger point techniques (strain/counterstrain technique and ischemic compression) versus other interventions such as neuromuscular technique, [52] ischemic compression plus stretching of the hamstring, [53] kinesiotaping [17] to reduce pain provoked by TrPs. Based on the qualitative analyses, no significant differences in pain intensity favoring manual trigger point therapies versus other passive treatments were found. Kinesiotaping was found to be potentially clinically relevant based on the minimal important difference found between groups (MD [95% CI] −1.30 cm [−2.05, −0.55]) [47] (Figure 2 and Table 2 (Effect size)).

### 3.4. Pressure Pain Threshold

**Comparison of manual trigger point therapy vs. control group:** One study [52] evaluated the effect of a manual trigger point technique (strain/counterstrain technique) compared to a control group (i.e., no treatment). The forest plot (Figure 2) and Table 2 (effect sizes) showed that the results favored manual trigger point therapy (MD [95%]: 0.70 kg/cm^2^ [0.48, 0.92]). However, this difference was not clinically relevant (Figure 2) [49].

**Comparison of manual trigger point therapy vs. other therapies:** Two studies [52,53] evaluated the effect of manual trigger point techniques (strain/counterstrain technique, ischemic compression) versus other intervention methods such as neuromuscular technique [52], or hamstring stretching [53]. There were no significant differences between groups for pressure pain threshold values in both studies (Table 1 and Figure 2) [49].

### 3.5. Maximum or Vertical Mouth Opening

**Comparison of manual trigger point therapy vs. control group**: Two studies [51,52] evaluated the effect of the manual trigger point technique (strain/counterstrain technique) compared to a control group (Figure 2, 2.3.6). Results of these studies were mixed; one study [51] found no statistical difference between groups and the other [52] found significant results favoring the manual trigger point therapy group. This study showed a clinically relevant increase on maximal mouth opening in the manual trigger point therapy group compared to the control group (MD [95% CI]: 6.00 mm [1.97, 10.03]) (Figure 2).

**Comparison of manual trigger point therapy vs. other therapies**: Three studies [51,52,53] evaluated the effect of strain/counterstrain technique or ischemic compression versus other intervention methods such as post-isometric relaxation [51], neuromuscular technique [52], or hamstring stretching [53]. None of the studies showed a clear and significant difference between manual trigger point therapy and any of the other techniques analyzed (Figure 2 and Table 1).

### 3.6. Risk of Bias of Analyzed Studies

The risk of bias (RoB) is summarized in detail in Appendix E and Figure 3. All the included studies for this systematic review used a random sequence generation, provided well defined inclusion and exclusion criteria for their populations, and gave a detailed description of the protocols used for treatment. However, all of them had some methodological flaws (Appendix E). Questions regarding the interventions revealed that the treatment protocol was inadequately described by all included studies, as some details of exposure were not reported. Therefore, the reproducibility of these interventions is not possible.

Regarding all aspects of the compiled set of items and the RoB assessment tool, the four studies included in this systematic review were considered to have an overall high risk RoB (Figure 3 and Figure 4, and Appendix E).

### 3.7. Quality of Evidence

The study quality was assessed using the GRADE approach [50]. The overall quality of evidence was very low (Table 2) due to the high RoB of all included studies and due to the indirectness in some of the comparisons in this systematic review. The evidence was generally downgraded for three reasons including RoB, imprecision, and indirectness of the reported results. 

## 4. Discussion

The findings from this systematic review show that manual trigger point therapy could potentially be beneficial for patients with mTrPs in the orofacial area. Even though manual trigger point therapy did not show a clear advantage over other conservative treatments, it was found to be an effective therapy for patients with mTrPs in the orofacial area and better than control groups. However, the number of studies was limited, the RoB was high, and the certainty of evidence was low. Despite the poor quality of the evidence, the non-invasive nature of the manual myofascial techniques can be an attractive complement or even alternative to other interventions. In addition, although we aimed to pool studies, this was not possible due to the heterogeneity of the results of the included studies.

### 4.1. Effectiveness of Manual Trigger Point Therapy in Comparison with Other Reviews

This systematic review results are in agreement with other systematic reviews focusing on other regions of the body, such as headache or neck pain, and myofascial pain syndromes [24,25,56]. All reviews agree that treatment with manual trigger point therapy techniques leads to a significant or promising improvement in several outcomes when compared with home exercises [57], physical therapy modalities (i.e., hot packs, transcutaneous electric nerve stimulation (TENS), stretch with spray, and others) [58], or transverse friction massage [59], but the level of evidence remains low to very low. Although previous reviews on this topic are available, they are already outdated (in the view of the Cochrane collaboration), as they were published more than 5 years ago, their conclusions can be maintained as their results agree with ours, which provides an updated analysis of this literature.

One finding of clinical importance from these reviews is that the included trials in these reviews had a low number of sessions or they did not implement follow-ups of the manual therapy treatment, similar to the findings of this systematic review. It is unlikely that short treatment exposure from these manual TrPs techniques (i.e., a limited number of sessions) can produce a significant and long-lasting effect as highlighted by our review.

### 4.2. Methodological Biases and Evidence Quality

As shown previously, all the included studies were considered to have a high risk of bias (Figure 3). These common biases in the included studies could have impacted the results of this review and are shown in Figure 4. The lack of blinding might have influenced the results of the analyzed studies, but due to the nature of the treatment methods used, blinding might not have been possible in all cases. There are some strategies that have been suggested to overcome lack of blinding in this type of studies that could be used in future research in this area [60]. Even though the studies reported the interventions they used adequately for their treatment groups, they did not report on co-interventions, adverse effects, or adequate adherence to the treatments. In addition, it is unclear whether the participants received all scheduled treatment sessions and whether they received a sufficient dose of treatments, especially since some of the included studies [51,53] used only one session of manual trigger point therapy (Table 1 and Table 2). Furthermore, none of the trials investigated any short-term or long-term effects of the interventions (no follow-ups). All these shortcomings make the evidence from these studies uncertain and poor.

### 4.3. Future Directions

This review shows that there is a paucity of studies looking at the effectiveness of manual trigger point therapy in individuals with myofascial trigger points in the orofacial region. In addition, from the few studies included, no high-quality evidence could be found, which indicates that there is great uncertainty about the effectiveness of manual trigger point therapy in comparison with other therapies in this area. There is a great need for well-designed RCTs considering the limitations highlighted by this review (see section above) to specifically investigate manual trigger point therapy for patients with mTrPs in the orofacial area [23,24,25].

### 4.4. Clinical and Research Implications

One of the striking results of this systematic review was that manual trigger point therapy techniques were only applied in a few sessions and were not followed up. Further studies are needed to examine the effect of manual trigger point therapy involving a longer period (more sessions) and measure the long-term effects of these interventions, especially the ischemic compression and strain/counterstrain techniques.

Surprisingly, none of the studies included in this systematic review used a placebo group. Thus, it is unclear, whether manual trigger point therapy was better than a placebo treatment. Future studies evaluating the effect of manual trigger point therapy techniques might need a stronger study design including a placebo intervention.

### 4.5. Strengths and Limitations of This Review

This systematic review has some strengths and limitations that need to be addressed at this point. The literature searches were performed by an experienced health sciences librarian. This enabled accurate identification of possible studies from the respective databases. In addition, our systematic review did not limit on the basis of language and we searched from 1946 to April 2021 (i.e., database inception to date of last search). Our inclusion criteria were also broad since our objective was to include all studies looking at manual trigger point therapies. Despite these broad criteria and the thorough search conducted, only four studies were found that met our criteria for inclusion. Furthermore, this review was the first to specifically investigate manual trigger point therapy in patients with latent mTrPs in muscles of the orofacial region. Nevertheless, the most common condition reported on was TMD. 

In addition, it is important to note that a high risk of bias was found across all included studies, which limits the confidence of the effect of manual trigger point therapies. In addition, due to the paucity of the evidence (i.e., limited number of studies) as well as the heterogeneity of the literature (i.e., not used standardized protocols in the studies), no meta-analysis of the effect estimates could be performed. In addition, we could not explore publication biases through a funnel plot due to the limited number of studies. However, due to the comprehensive searches performed, we do not believe this is a concern.

## 5. Conclusions

The results from this systematic review support that strain/counterstrain therapy was superior to control groups for patients with mTrPs in the orofacial area to improve pain intensity, pain pressure threshold and mouth opening. However, manual trigger point therapy was equivalent to other active treatment techniques. Overall, the quality of evidence was very low and the risk of bias was high. Therefore, manual trigger point techniques could potentially be used as a complementary technique in the treatment of patients with mTrP in the orofacial area. In addition, there is a paucity of well conducted RCTs in patients with mTrPs in the orofacial region. Rigorous, well-designed RCTs are still needed in this field.

## Figures and Tables

**Figure 1 life-13-00336-f001:**
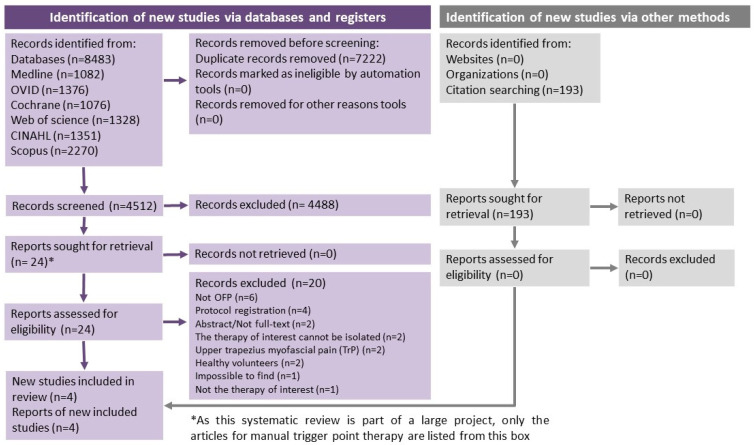
Flow diagram of included studies for this systematic review based on PRISMA guidelines.

**Figure 2 life-13-00336-f002:**
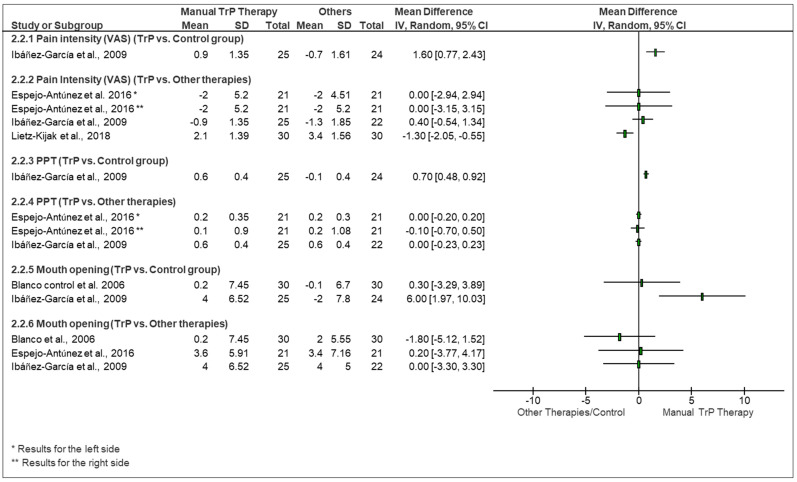
Forest plot for qualitative comparison of manual TrP therapy vs. any other therapy or controls used in the analyzed studies. Pain is reported in cm, pressure pain thresholds (PPT) in kg/cm^2^, and mouth opening in mm).

**Figure 3 life-13-00336-f003:**
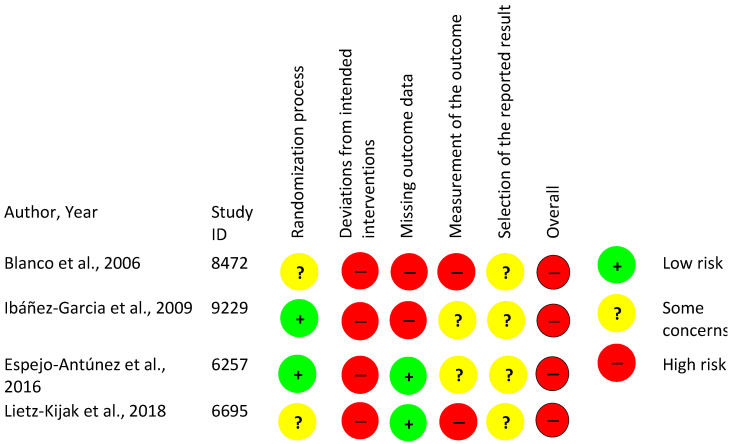
Risk of bias summary performed using the Cochrane Risk of Bias tool to evaluate the quality of the RCTs.

**Figure 4 life-13-00336-f004:**
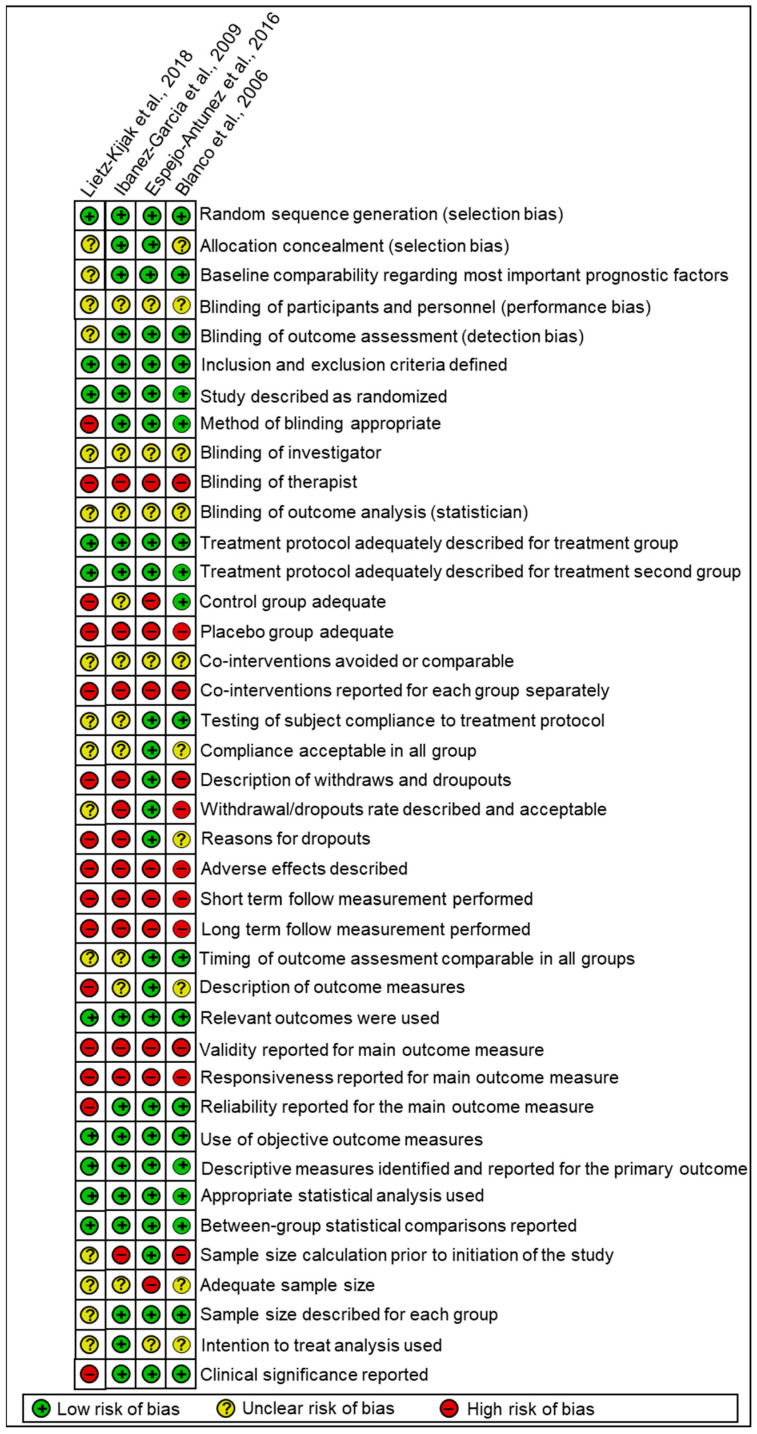
Risk of bias summary performed using the compiled set of items to evaluate the quality of the RCTs.

**Table 1 life-13-00336-t001:** Summary of randomized controlled trials included in this systematic review.

Study (Year)	Intervention vs. Comparison Group(s)	Outcome Measure—Measure Tool	Results between Groups MD [95% CI]	Results between Groups *p* Value and Clinical Significance Assessment	Conclusion
Blanco et al. (2006) [51]	SC (masseter) vs.PIR (masseter) vs. Control: no therapy	MMO (mm)—NR	**MMO**: **SC vs. PIR** −1.80 mm [−5.12, 1.52] **SC vs. CO** 0.30 mm [−3.29, 3.89]	**MMO**: **SC vs. PIR** (*p* = 0.4090; no significant difference between groups; NCS) **SC vs. CO** (*p* = 0.84, no significant difference between groups; NCS)	For participants with latent mTrPs in the masseter muscle, the post-isometric relaxation technique showed a greater effect on active mouth opening than the strain/counterstrain technique.
Ibáñez-García et al. (2009) [52]	SC (masseter) vs. NT (masseter) vs. Control: no therapy	Pain intensity (0–10 cm)—VAS PPT (kg/cm^2^)—mechanical pressure algometer MMO (mm)—NR	**Pain intensity**: **SC vs. NT** 0.40 cm [−0.54, 1.34] **SC vs. CO** 1.60 cm [0.77, 2.43] **PPT**: **SC vs. NT** 0.0 kg/cm^2^ [−0.23, 0.23] **SC vs. CO** 0.70 kg/cm^2^ [0.48, 0.92] **MMO**: **SC vs. NT** 0.00 mm [−3.30, 3.30] **SC vs. CO** 6.00 mm [1.97, 10.03]	**Pain intensity**: **SC vs. NT** (*p* = 0.9, no significant difference between groups; NCS) **SC vs. CO** (*p* <0.001, significant difference favoring SC; PCSR) **PPT**: **SC vs. NT** (*p* = 0.9, no significant difference between groups; NCS) **SC vs. CO** (*p* < 0.001, significant difference favoring SC; NCS) **MMO**: **SC vs. NT** (*p* = 0.9, no significant difference between groups; NCS) **SC vs. CO** (*p* < 0.001, significant difference favoring SC; CSR)	The neuromuscular or strain/ counterstrain technique showed in the results’ increased PPTs, increased active mouth opening, and decreased local pain by pressure over latent myofascial TrPs in the masseter muscle. For both intervention groups, the large effect sizes suggest a strong clinical effect, while the effect size of the control was small.
Espejo-Antúnez et al. (2016) [53]	IC (masseter) + HS (hamstrings) vs. HS (hamstrings)	Pain intensity (0–10 cm) – VAS PPT (kg/cm^2^) – mechanical pressure algometer MMO/VMO (mm) – calibrated caliper ROM (degree °) – computer analysis of photographs	**Pain intensity**: **HS vs. HS+IC_R** 0.00 cm [−3.15, 3.15] **HS vs. HS+IC_L** 0.00 cm [−2.94, 2.94] **PPT**: **HS vs. HS+IC_R** −0.10 kg/cm^2^ [−0.70, 0.50] **HS vs. HS+IC_L** 0.00 kg/cm^2^ [−0.20, 0.20] **MMO/VMO**: **HS vs. HS+IC** 0.20 mm [−3.77, 4.17]	**Pain intensity**: **HS vs. HS+IC_R** (*p =* 1.0; no significant difference between groups; NCS) **HS vs. HS+IC_L** (*p* = 1.0; no significant difference between groups; NCS) **PPT**: **HS vs. HS+IC_R** (*p* = 0.7616; no significant difference between groups; NCS) **HS vs. HS+IC_L** (*p* = 1.0; no significant difference between groups; NCS) **MMO/VMO**: **HS vs. HS+IC** (*p* = 0.4708; no significant difference between groups; NCS)	Both groups showed an increased hamstrings extensibility, active mouth opening, and pressure pain threshold, as well as decreased pain intensity. Adding ischemic compression did not result in further improvements in hamstrings extensibility or clinical features of TMD.
Lietz-Kijak et al. (2018) [17]	IC (masseter) vs. KT (masseter)	Pain intensity (0–10 cm) – VAS	**Pain intensity**: **KT vs. IC** −1.30 cm [−2.05, −0.55]	**Pain intensity**: **KT vs. IC** (*p* < 0.001; significant difference favoring KT; PCSR)	Significant analgesic effects were achieved by kinesiotaping (KT) and TrP inactivation for the treatment of painful forms of functional disorders of the masticatory muscles; however, more beneficial results were observed in the KT group.

CSR: clinically significant result; CO: Control group; d: days; F: female; HS: Hamstring-stretching; IC: Ischemic compression; SC: Strain/Counterstrain technique; KT: Kinesiotaping; L: left side; M: male; MTT: Manual TrP therapy; MMO: Maximal mouth opening; MD: Mean difference; m: months; NA: Not applicable; NCS: not clinically significant; NR: not reported; NT: Neuromuscular technique; PIR: Post-isometric relaxation; PCSR: potentially clinically significant result; PPT: Pressure pain threshold; ROM: Range of motion (knee extension); R: right side; TT: time of treatment of mTrP; T: treatment; w: week; VMO: vertical mouth opening.

**Table 2 life-13-00336-t002:** Quality of evidence (GRADE) between manual TrP therapy versus other interventions.

	Quality Assessment						Summary of Findings			
							No. of Patients		Effect	
Comparisons	No. of Studies (Design; Time of measurement)	Risk of Bias	Inconsis-tency	Indirectness	Imprecision	Publication Bias	Patients in Tx group	Patients in Comparison/ Control Group	Estimate (MD) [95% CI]	Quality
**Pain Intensity (assessed with VAS: scale from 0 – 10 cm)—Manual TrP therapy vs. Control**										
Strain/Counterstrain technique vs. Control (no treatment)	1 RCT [52]; right after treatment	Very Serious ^(a)^	NA	Serious ^(d)^	Serious ^(e)^	Undetected ^(f)^	25	24	MD = 1.60 cm [0.77, 2.43]	**Low**
**Pain Intensity (assessed with VAS: scale from 0 – 10 cm)—Manual TrP therapy vs. Other therapies**										
Hamstring stretching + Ischemic compression vs. Hamstring stretching (left)	1 RCT [53]; right after treatment	Very Serious ^(a)^	NA	No serious ^(c)^	Serious ^(e)^	Undetected ^(f)^	21	21	MD = 0.00 cm [−2.94, 2.94]	**Low**
Hamstring stretching + Ischemic compression vs. Hamstring stretching (right)	1 RCT [53]; right after treatment	Very Serious ^(a)^	NA	No serious ^(c)^	Serious ^(e)^	Undetected ^(f)^	21	21	MD = 0.00 cm [−3.15, 3.15]	**Low**
Strain/Counterstrain technique vs. Neuromuscular technique	1 RCT [52]; right after treatment	Very Serious ^(a)^	NA	Serious ^(d)^	Serious ^(e)^	Undetected ^(f)^	22	22	MD = 0.40 cm [−0.54, 1.34]	**Low**
Ischemic compression vs. Kinesio taping	1 RCT [17]; right after treatment	Very Serious ^(a)^	NA	No serious ^(c)^	Serious ^(e)^	Undetected ^(f)^	30	30	MD = −1.30 cm [−2.05, −0.55]	**Low**
**Pressure Pain Threshold (assessed with pressure algometer: kg/cm^2^)—Manual TrP therapy vs. Control**										
Strain/Counterstrain technique vs. Control (no treatment)	1 RCT [52]; right after treatment	Very Serious ^(a)^	NA	Serious ^(d)^	Serious ^(e)^	Undetected ^(f)^	25	24	MD = 0.70 kg/cm^2^ [0.48, 0.92]	**Low**
**Pressure Pain Threshold (assessed with pressure algometer: kg/cm^2^)—Manual TrP therapy vs. Other therapies**										
Hamstring stretching + Ischemic compression vs. Hamstring stretching (left)	1 RCT [53]; right after treatment	Very Serious ^(a)^	NA	No Serious ^(c)^	Serious ^(e)^	Undetected ^(f)^	21	21	MD = 0.00 kg/cm^2^ [−0.20, 0.20]	**Low**
Hamstring stretching + Ischemic compression vs. Hamstring stretching (right)	1 RCT [53]; right after treatment	Very Serious ^(a)^	NA	No Serious ^(c)^	Serious ^(e)^	Undetected ^(f)^	21	21	MD = −0.10 kg/cm^2^ [−0.70, 0.50]	**Low**
Strain/Counterstrain technique vs. Neuromuscular technique	1 RCT [52]; right after treatment	Very Serious ^(a)^	NA	Serious ^(d)^	Serious ^(e)^	Undetected ^(f)^	25	22	MD = 0.00 kg/cm^2^ [−0.23, 0.23]	**Low**
**Mouth Opening (assessed with calibrated caliper (Blanco et al.): scale in mm; in Ibáñez-García: NR: scale in mm) – Manual TrP therapy vs. Control**										
Strain/Counterstrain technique vs. Control (no treatment)	1 RCT [51]; right after treatment	Very Serious ^(a)^	Very serious ^(b)^	Serious ^(d)^	Serious ^(e)^	Undetected ^(f)^	30	30	Total MD = 0.30 mm [−3.29, 3.89]	**Very Low**
Strain/Counterstrain technique vs. Control (no treatment)	1 RCT [52]; right after treatment	Very Serious ^(a)^	Very serious ^(b)^	Serious ^(d)^	Serious ^(e)^	Undetected ^(f)^	30	30	Total MD = 6.00 mm [1.97, 10.03]	**Very Low**
**Mouth Opening (assessed with calibrated caliper (Espejo-Ant** **únez et al.): scale in mm; in Ibáñez-García: NR: scale in mm)—Manual TrP therapy vs. Other therapies**										
Strain/Counterstrain technique vs. post-isometric relax.	1 RCT [51]; right after treatment	Very Serious ^(a)^	NA	Serious ^(d)^	Serious ^(e)^	Undetected ^(f)^	30	30	MD = −1.80 mm [−5.12, 1.52]	**Very Low**
Hamstring stretching + Ischemic comp. vs. Hamstring stretching	1 RCT [53]; right after treatment	Very Serious ^(a)^	NA	No serious ^(c)^	Serious ^(e)^	Undetected ^(f)^	21	21	MD = 0.20 mm [−3.77, 4.17]	**Low**
Strain/Counterstrain technique vs. Neuromuscular technique	1 RCT [52]; right after treatment	Very Serious ^(a)^	NA	Serious ^(d)^	Serious ^(e)^	Undetected ^(f)^	25	22	MD = 0.00 mm [−3.30, 3.30]	**Very Low**

CI: Confidence Interval; MD: Mean difference; NA: Not applicable. **Explanations: ^(a)^** Allocation concealment unclear in two studies ([17,51], performance bias and co-interventions are unclear, and incomplete outcome data have a high risk of bias for all four studies. Three studies with unclear intention to treat [17,51,53]. All studies were considered to have high risk of bias overall. **^(b)^** Both studies used control groups without receiving any treatment, and the same treatment method for manual TrP therapy (strain/ counterstrain technique). Both studies investigated the effect of manual TrP on patients with latent myofascial TrPs in the masseter muscle. However, a high heterogeneity was obtained in the effect estimates. **^(c)^** No serious indirectness, as these studies match the PICO format of this systematic review. **^(d)^** Serious indirectness, as these studies do not match with all aspects of PICO format; population with latent myofascial TrPs. **^(e)^** All the studies have a wide confidence interval (CI) and unclear or inadequate sample sizes. **^(f)^** Undetected publication bias due to the precise literature search performed by an experienced scientific librarian.

## Data Availability

Data can be made available upon request to the authors.

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
