# Peer review of "Effectiveness of Manual Trigger Point Therapy in Patients with Myofascial Trigger Points in the Orofacial Region—A Systematic Review"

_life, 2023, doi:10.3390/life13020336_

Round 1

Reviewer 1 Report

Dear Authors,

this is an interesting piece of reading. However, I have some minor concerns:

1) the descriptions of the tables and figures must be revised as, for instance, it is not needed to explain in the title of the figure the strategy of searching

2) lines 235,236 - you submitted a systematic review, so why do you mention that meta-analysis was not possible to be prepared?

3) I would highly recommend to use the alphabetic order of the abbreviations in the footer.

4) the synthesis of results should be rewritten to make it more clear for the reader

5) the first sentence in the conclusions does not support the previously described findings - the conclusions must summarize the most important findings from the manuscript

6) English correction is needed.

I accept this manuscript after minor revision.

Author Response

All concerns from the previous review process have been carefully and fully addressed in the preparation of this revised version of the manuscript. Each of these concerns has been addressed below and requested changes have been made to the manuscript. Below is a list of reviewers’ requests and the changes made by our team.

Thank you for your effort and time,

1) Thank you for your comment. All the tables´ titles were reviewed and cleaned to keep the title concise without further descriptions.

2) Initially, the idea of this study was to perform a meta-analysis, however, due to the number of studies and heterogeneity among them, this was not possible. This was mentioned in our PROSPERO protocol. Therefore, we would like to keep this statement to explain the difference between the protocol and the final manuscript. We also want to clarify that Figure 2 is not a meta-analysis; it is a forest plot, which displays the direction of the effect estimates of the included studies.

3) Thank you for the suggestion. Tables and figures´ footnotes were organized in alphabetic order as suggested.

4) We appreciate the comment. The paragraph has been rewritten to make our results clearer for the reader.

5) Thanks for bringing this to our attention. We have changed the text to describe the results more precisely and focus on the main results as suggested.

6) Thank you for your comment. The manuscript has now been fully reviewed by a native English-speaker

Reviewer 2 Report

Dear Author,

This systematic review aimed to synthesize and evaluate the quality of the evidence from RCTs or clinical trials regarding the effectiveness of manual trigger point therapy compared with other treatment strategies for managing mTrPs in the orofacial area in individuals with or without orofacial pain.

Abstract

https://www.mdpi.com/journal/life/instructions

- Please follow the Instruction for Authors. 

  • Abstract: The abstract should be a total of about 200 words maximum. The abstract should be a single paragraph and should follow the style of structured abstracts, BUT WITHOUT HEADINGS: 1) Background: Place the question addressed in a broad context and highlight the purpose of the study; 2) Methods: Describe briefly the main methods or treatments applied. Include any relevant preregistration numbers, and species and strains of any animals used. 3) Results: Summarize the article's main findings; and 4) Conclusion: Indicate the main conclusions or interpretations. The abstract should be an objective representation of the article: it must not contain results which are not presented and substantiated in the main text and should not exaggerate the main conclusions.

- Please remove headings.

- Please remove 

Simple Summary: Problem statement: Myofascial trigger points are hypersensitive or tender spots 25 located in the muscles, when compressed or stretched can cause referred or local pain. Aim: 1) To 26 evaluate studies that investigated the effectiveness of manual trigger point therapy for treat myo-27 fascial trigger points in the orofacial area in individuals with or without orofacial pain, and thus 28 inform future practice and provide recommendations regarding manual trigger point therapy to 29 treat this population. Results: Four studies were included and evaluated in this review. The quality 30 of the evidence was very low, and there was a high risk of bias on the included studies. Manual 31 trigger point therapy did not show a clear advantage over other conservative treatments; however, 32 it was better than no treatment to treat patients with myofascial trigger points in the orofacial area. 33 Thus, this therapy can be used as a complement or alternative therapy to other interventions. Con-34 clusions: Although the quality of the overall evidence is very low, the results from this systematic review 35 showed positive results using manual trigger point therapy to treat myofascial trigger points in the 36 orofacial region when compared with no treatment.”

- Line 40: Please remove “or without”

- “Since manual trigger point therapy did not 45 show a clear advantage over other conservative treatments, it was found to be equally effective and 46 a safe therapy for individual with myofascial trigger points in the orofacial region and better than 47 control groups.” This sentence was not clear.

Introduction

I suggest improving the introduction section, which is poor and should be entirely modified.

·      After definition, please report the classification the OFP (International Classification of Orofacial Pain (ICOP)) (International Classification of Orofacial Pain, 1st edition (ICOP). Cephalalgia, 2020. 40(2): p. 129-221.).

·      Moreover, improve epidemiological data, reporting that temporomandibular disorder is the second most common musculoskeletal disorder that causes pain and disability and cite the most recent literature (Prevalence of temporomandibular joint disorders: a systematic review and meta-analysis. Clin Oral Investig. 2021 Feb;25(2):441-453. doi: 10.1007/s00784-020-03710-w. Epub 2021 Jan 6. PMID: 33409693; Global burden of oral diseases: emerging concepts, management and interplay with systemic health. Oral Dis 2016; 22(7):609-19. Doi: 10.1111/odi.12428).

·      Then, report the commonly used treatments for myogenous TMD (discuss and cite. Efficacy of rehabilitation on reducing pain in muscle-related temporomandibular disorders: A systematic review and meta-analysis of randomized controlled trials. J Back Musculoskelet Rehabil. 2022;35(5):921-936. doi: 10.3233/BMR-210236. Effects of Occlusal Splints on Spinal Posture in Patients with Temporomandibular Disorders: A Systematic Review. Healthcare (Basel). 2022 Apr 15;10(4):739. doi: 10.3390/healthcare10040739.). These recent SRs with meta-analysis include TMD patients and include paper evaluating the efficacy of physical therapy and dry needling and occlusal splint, also in relation to the posture.

·      So, report the clinical manifestations as pain, decrease in the mouth opening, muscle or joint tenderness on palpation, limitation of mandibular movements, joint sounds and otologic complaints like tinnitus, vertigo or ear fullness, etcetera.

·      Please, better define the rationale of the study.

·      Diagnosis and Etiology

·      In my opinion it was very important to report that the most widely used methods to assess skeletal maturation, as the hand and wrist rx as gold standard, and the CVM by Baccetti. Two systematic reviews were conducted on this topic in the last years, and authors concluded that the CVM method shows a high level of correlation with the HWM method. Please refer to and cite: Szemraj et al. Is the cervical vertebral maturation (CVM) method effective enough to replace the hand-wrist maturation (HWM) method in determining skeletal maturation?-A systematic review. Eur J Radiol. 2018 May;102:125-128. doi: 10.1016/j.ejrad.2018.03.012. and Ferrillo et al. Reliability of cervical vertebral maturation compared to hand-wrist for skeletal maturation assessment in growing subjects: A systematic review. J Back Musculoskelet Rehabil. 2021;34(6):925-936. doi: 10.3233/BMR-210003.

Materials and Methods

- Matherial and methods were very correctly reported.

- I suggest to modify Figure 1 reporting the PRISMA 2020 flow diagram for updated systematic reviews (Page MJ, McKenzie JE, Bossuyt PM, Boutron I, Hoffmann TC, Mulrow CD, et al. The PRISMA 2020 statement: an updated guideline for reporting systematic reviews. BMJ 2021;372:n71. doi: 10.1136/bmj.n71. For more information, visit: http://www.prisma-statement.org/).

Discussion 

Very well conducted.

Report the References according to 

https://www.mdpi.com/journal/life/instructions

Author Response

All concerns from the previous review process have been carefully and fully addressed in the preparation of this revised version of the manuscript. Each of these concerns has been addressed below and requested changes have been made to the manuscript. Below is a list of reviewers’ requests and the changes made by our team.

Thank you for your effort and time,

1) The Abstract was modified as requested; headings were removed. 

2) Simple summary is one of the requests from the journal. Therefore, we would like to keep it. It was not specified on the website; however, it was one of the topics in the template that was provided to us.

3) We would like to retain this phrase because some of the selected studies for this systematic review reported that patients had mTrPs in the orofacial area (active or latent) but the diagnosis of orofacial pain was not present in all individuals. Therefore, it is more accurate to state that the subjects included in these studies had mTrPs with or without OFP.

4) We have edited this sentence to facilitate its understanding as follows: “Manual trigger point therapy showed no clear advantage over other conservative treatments. However, it was found to be an equally effective and safe therapy for individuals with myofascial trigger points in the orofacial region and better than control groups.”

5) 

We could not understand clearly what the request from the reviewer was. We have added the classification of OFP and its reference in the section population. In this section, we described the population targeted in this review, and OFP was one of the diagnoses of interest.

The ICOP classification is very extensive. We have added an appendix with the diagnoses from this classification included in this review, making it easier for the reader to have this information.

We also added the definition of OFP and added the requested reference in the introduction.

5) Thanks for these suggestions. We have added the information suggested by the reviewer. References to these studies were also added. However, one of the reviews suggested by the reviewer focused on the prevalence of TMJ joint disorders.[1] Since we did focus on Myofascial trigger points (mTrPs), we focused on their prevalence as suggested by the literature.

6) Thanks for the suggested literature. We have added a paragraph highlighting the general treatments for orofacial pain and specifically TMD and added the suggested and other references to support our statements.

7) We have added this information in the introduction as suggested.

8) We have edited the last paragraph of the introduction to highlight the weaknesses of existing reviews and identify the gap in the literature.

9) Figure 1 already follows the new PRISMA 2020.

10) Thank you.

11) 

The style according to the journal rules is free as stated in the guidelines for authors: “Your references may be in any style, provided that you use the consistent formatting throughout. It is essential to include author(s) name(s), journal or book title, article or chapter title (where required), year of publication, volume and issue (where appropriate) and pagination. DOI numbers (Digital Object Identifier) are not mandatory but highly encouraged. The bibliography software package EndNote, Zotero, Mendeley, Reference Manager are recommended.”

However, in the template provided for this issue, references between square brackets have been suggested.

Round 2

Reviewer 2 Report

Authors modified the text according to the suggestions.

I found this work impactful and it fits well with in the scope of this journal.

In my opinion, it is suitable for publication.